# *In silico* analysis and experimental validation shows negative correlation between miR-1183 and cell cycle progression gene 1 expression in colorectal cancer

**Syeda Alina Fatima**[1⊙]**, Mubeen Tabish Nasim**[1⊙]**, Ambrin Malik**[1]**, Saif Ur Rehman**[1]**, Saboora Waris**[2]**, Manal Rauf**[3]**, Syed Salman Ali**[4,5]**, Farhan Haq**[1,6]**, Hassaan Mehboob Awan**[1]*

1 Department of Biosciences, Cancer Genetics and Epigenetics Lab, COMSATS University Islamabad, Islamabad, Pakistan, 2 Maroof International Hospital, Islamabad, Pakistan, 3 Pakistan Institute of Medical Sciences, Islamabad, Pakistan, 4 Combined Military Hospital, Kharian, Pakistan, 5 Department of Cellular Pathology, Royal London Hospital, Barts Health, NHS Trust, London, United Kingdom, 6 Division of Microbiology, Immunology and Glycobiology, Lund University, Lund, Sweden

⊙ These authors contributed equally to this work.
* hassaan.awan@comsats.edu.pk

**Data Availability Statement:** All relevant data are within the manuscript and its Supporting Information files.

## Abstract

MicroRNAs (miRNAs) are small noncoding RNAs that post-transcriptionally regulate gene expression by binding to the 3' untranslated regions (UTR) of target genes. Aberrant expression of miRNAs can lead to disease, including cancer. Colorectal cancer (CRC) is one of the leading causes of cancer-related deaths worldwide. Among several factors, differential expression of miRNA can have serious consequences on disease progression. This study was designed to computationally identify and experimentally verify strong miRNA candidates that could influence CRC progression. *In silico* analysis of publicly available gene expression microarray datasets revealed significant upregulation of miR-1183 in CRC. Comparison of mRNA microarray expression data with predicted miR-1183 targets led to the identification of cell *cycle progression gene 1* (*CCPG1*) as strong, negatively correlated miR-1183 target. Expression analysis by means of quantitative PCR validated the inverse correlation between miR-1183 and *CCPG1* in colorectal cancer tissues. *CCPG1* indirectly modulates the cell cycle by interacting with the PH/DH domain of Dbs (Rho-specific guanine nucleotide exchange factor). Interestingly, the computational analysis also showed that miR-1183 is upregulated in liver and gastric cancer. This finding is notable as the liver and stomach are the primary metastatic sites for colorectal cancer and hepatocellular carcinoma respectively. This novel finding highlights the broader implications of miR-1183 dysregulation beyond primary CRC, potentially serving as a valuable prognostic marker and a therapeutic target for both primary and metastatic CRC.

**Funding:** The author(s) received no specific funding for this work.

# 1. Introduction

Small noncoding RNAs of ~22 nucleotides in length, known as miRNA, are important post-transcriptional gene regulators. They control essential biological processes including, but not limited to; cell division, proliferation, differentiation, apoptosis etc. [1]. Differential expression of miRNAs can be detrimental, leading to developmental defects and disease initiation and progression. The clinical understanding of miRNA expression in different cancers is the need of the hour. It will contribute to a better understanding of the disease, which may lead to the development of better treatments. For example, the dysregulation of miR-21 is involved in the initiation and progression of several cancers including breast, lung, liver, and colorectal cancer [2–5]. The same miRNA can be targeted by Curcumol, an anti-proliferative drug, as a treatment to inhibit cell proliferation in colorectal cancer [6]. Importantly, studies suggest that these dysregulated miRNAs can be used as potential targets for chemotherapeutic treatment [7].

Colorectal cancer (CRC) is one of the most common (second most common) malignancies with a high mortality rate worldwide [8]. Of the 19 million reported cancer cases worldwide, 1.9 million cases were colorectal cancer. Similarly, out of 10 million total cancer-related deaths in 2020, 0.94 million were deaths due to CRC [9,10]. Differential expression of miRNAs play important roles in tumorigenesis of CRC [11]. Despite the significant progress in identifying the role of microRNAs in CRC, clinical relevance of miRNAs in colorectal cancer remains elusive.

In this study, we carried out a comprehensive computational analysis of microarray data, which was obtained from four independent cohorts comprising a total of 3,074 colorectal cancer patients. Our analysis revealed significant upregulation of miR-1183 in all the CRC datasets analyzed, suggesting its potential involvement in CRC tumorigenesis. Furthermore, the expression of *cell cycle progression gene 1* (*CCPG1*) negatively correlated with miR-1183 expression, suggesting a potential regulatory relationship. The differential expression and inverse correlation between the two genes was experimentally verified using RT-qPCR data. Interestingly, miR-1183 was upregulated in hepatocellular carcinoma (HCC), and gastric cancer (GC). The aberrant expression of miRNA was observed by differential expression analysis of microarray data and RT-qPCR. The result showed that this miRNA potentially targets the *CCPG1*. Targeting miR-1183 Targeting miR-1183 may play a key role in the prognosis of CRC, which is a novel discovery and is of increasing importance in the field of oncogenesis., This study is limited to only a small sample size, and to a specific population. As the increased expression of miR-1183 and decreased expression of *CCPG1* have not been previously discussed together in CRC. The novel finding underscores the broader implications of miR-1183 dysregulation beyond primary CRC, potentially serving as a valuable prognostic marker and a therapeutic target for both primary and metastatic CRC.

# 2. Methodology

## 2.1. Data collection

Publicly available miRNA microarray datasets (GSE41655, GSE30454, GSE39814, GSE106817) for colorectal cancer (CRC) comprising of 3.704 samples, gastric cancer (GC) dataset (GSE106817, comprising of 2,876 samples) and hepatocellular cancer (HCC) dataset (GSE106817, comprising of 2,842 samples), were downloaded from GEO NCBI [12] and ArrayExpress [13] databases. The GC and HCC datasets were also retrieved as we were interested in analyzing if there is any miRNA that is commonly playing a role in all these three cancers. Both (GEO NCBI and ArrayExpress) browsers have publicly available functional genomics datasets from array as well as sequencing. Dataset information is provided in S1

Table in S1 File. All the relevant datasets were browsed and only those were selected that contained: microarray data, both normal and tumor samples, belonged to organism *Homo Sapiens*, had noncoding RNA data, and those having tumor stage information available. The other datasets that contained only tumor samples, or expression data from cell culture experiments, or those belonged to any other organism such as *Mus musculus*, or in a language other than English was excluded from this research.

## 2.2. Dataset normalization and Identification of differentially expressed miRNAs

Microarray datasets were analyzed through GEO2R [12], which is a web tool used for the comparing of samples. Samples were classified according to their respective tissue types i.e., normal and tumor tissues. R-version 3.6.3 was used for the analysis. Data was normalized using Robust Multichip Average (RMA) normalization method. Limma, and GEOquery library was used for differential expression, normalization, and $\log_2$ transformation. The analysis was conducted by GEO2R in an automated way after the selection of two groups i.e., tumor versus normal tissue samples.

Clinical information such as age, gender, pathologic grade and pathological tumor-node-metastasis (pTNM) stages were used to identify differentially expressed miRNAs. Heat map was generated using Heatmapper [14] to identify differentially expressed miRNAs in CRC1 dataset. Differentially expressed common miRNAs were identified through Venn diagram (http://bioinfogp.cnb.csic.es/tools/venny/index.html). Cytoscape v3.9.1 was used for the construction of network diagrams [15].

## 2.3. Identification of miRNA target genes and their correlation with miRNA

Gene expression profiling microarray dataset (GSE41657) titled "Gene expression profiling of colorectal normal mucosa, adenoma and adenocarcinoma tissues" was used for this purpose. Correlation analysis was carried out between mRNA and miRNA expression data to find out a potential negatively correlated target. This was done through IBM SPSS® software. Potential miRNA target genes were identified by comparing; the predicted targets from TargetScan [16], miRmap [17] with these negatively correlated genes. The functionality of these genes and their role in biological processes were obtained from NCBI and Ensembl databases [12,18].

## 2.4. Sample collection

For this study, fifty formalin-fixed paraffin-embedded (FFPE) biopsy tissue blocks were collected from Pakistan Institute of Medical Sciences (PIMS), Islamabad. Similarly, fifty control samples were also collected. The archived samples and medical records were collected for the year range 2016–2019. The data and samples obtained were anonymized and the patient's name, ID, date of birth, and contact number was hidden. The samples were verified by histopathologist and clinicopathological data was saved against each verified sample. Prior approval was sought from the ethical review board of COMSATS University Islamabad (#CUI/Bio/ERB/2021/53).

## 2.5. RNA extraction

The RNA was extracted from tissues preserved in FFPE blocks using the protocol described in [19]. Extraction of RNA was carried out on fifty pre-existing FFPE blocks of tumor tissues along with the fifty control samples. Using TRIzol Reagent (Invitrogen, Carlsbad, CA, USA),

total RNA was isolated, and its concentration was determined using a nanodrop spectrophotometer (NanoPhotometer Pearl, IMPLEN, Munich, Germany) while considering the samples with a 260/280 absorbance ratio of 2.0 or below.

### 2.6. cDNA synthesis

RNA extraction was followed by cDNA synthesis. cDNA was synthesized using RevertAid First Strand cDNA Synthesis Kit (Thermo Fisher Scientific) as per the manufacturers' instructions. Thermocycler profile was set at (25˚C for 5 min, 42˚C for 60 min, 85˚C for 5 min and, 4˚C for ~). In each reaction mixture, oligo dT primers along with miR-1183 stem-loop (SL) primers were added (S2 Table in S1 File).

### 2.7. Primer designing

Stem-loop primers for candidate miRNA was designed as described in [20]. Forward and reverse primers for miRNA and target genes were designed using Integrated DNA Technology software (IDT, San Diego, CA. USA). The oligos used in this study are given in S2 Table in S1 File. U6 was used as internal control for gene expression analysis.

### 2.8. qPCR

The VeriQuest SYBR Green qPCR Master Mix (Thermo Fisher Scientific, CA, United States) was used for quantitative real-time PCR (qRT-PCR). The miR-1183 and its target gene *CCPG1* expression were normalized using U6 as an internal reference. The reaction conditions comprised of an initial denaturation at 95˚C for 8 minutes, followed by 35 cycles of denaturation at 95˚C for 30 seconds, annealing at 58˚C, extension at 72˚C for 42 seconds and final extension at 72˚C for 15 minutes. The Livak's method ($2^{-\Delta\Delta Ct}$) was used to assess the relative expression and fold change.

## 3. Results

### 3.1. Clinical features of colorectal cancer dataset

Colorectal cancer (CRC) dataset 1 (CRC1) was comprised of total four types of CRC tissues including 15 normal mucosae, 39 low-grade adenomas, 20 high-grade adenomas and 33 adenocarcinomas. Colorectal cancer (CRC) dataset 2 (CRC2) contained information of 4 different groups that include the normal colonic mucosa, Lynch syndrome tumors, sporadic MSI tumors and MSS tumors. Moreover, it contained information regarding tissue type, disease state, immunohistochemistry, age, gender, race, tumor location and tumor stage. Colorectal cancer (CRC) dataset 3 (CRC3) contained information regarding miRNA source, biological source of exosomes of cell, disease state, cell type and cell line. In colorectal cancer (CRC) dataset 4 (CRC4), biological source information is given. Out of these four datasets, CRC1 and CRC2 contain the tumor stage information whereas; the other two datasets (CRC3 and CRC4) lack this information. The detailed clinical features of colorectal cancer (CRC) dataset 1 (CRC1) are given in S3 Table in S1 File.

### 3.2. Identification of differentially expressed miRNAs

In cohort 1, out of total 939 miRNAs extracted from CRC1, the differential expression of significantly upregulated miRNAs (107) and downregulated miRNAs (140) with p ≤ 0.05 were visualized using heat map (S1 Fig). The list of these 247 miRNAs with their logFC and p-values are given in (S4 Table in S1 File). The selection of miRNAs was narrowed down by considering

upregulated miRNAs with p ≤ 0.05 and logFC ≥ 1.5 as promising candidates for further analysis.

Out of 247, only 24 miRNAs were significantly upregulated with p ≤ 0.05 and logFC ≥ 1.5 (Figs 1 and S2A). Similarly, 30 miRNAs, out of 140, showed downregulation with p ≤ 0.05 and logFC ≤ -1.5 (Figs 1 and S2B).

The miRNAs were also plotted against their expression values in different tumor stages (S3 Fig). No direct relationship of miRNA expression values in different tumor stages was found. miRNAs were clustered based on their expression patterns in different T-stages. For

**Fig 1. Heat map of differentially expressed miRNAs in colorectal cancer dataset GSE41655 (CRC1) with logFC ≥ 1.5 (upregulated miRNAs) and logFC ≤ -1.5 (downregulated miRNAs).**

this study, we only focused on miRNAs that showed increased expression, as upregulated miRNAs in cancer generally behaves as oncogene, also known as oncomiRs.

## 3.3. Common upregulated miRNAs in colorectal cancer datasets

Multiple studies have shown miRNAs to be upregulated as well as downregulated in cancer progression [21,22]. In this study, we were particularly interested in identifying upregulated miRNAs that may play a crucial role in colorectal cancer. Therefore, three more datasets (CRC2, CRC3, and CRC4) were downloaded, normalized, sorted, and compared with each other to identify common miRNAs.

Upregulated miRNAs (logFC $\geq$ 1.5 and p-value $\leq$ 0.05) in CRC2, CRC3 and CRC4 collectively had a total of 1752 miRNAs. When CRC1 was compared with CRC2 (first validation cohort), a total of 4 miRNAs were common (Fig 2A). Comparison of CRC1 with CRC3 (second validation cohort) and CRC4 (third validation cohort) showed 17 (Fig 2B) and 5 overlapped miRNAs (Fig 2C), respectively. Collectively, only 2 miRNAs viz., has-miR-1183 and hsa-miR-630 were common in all four datasets (Fig 2D). The overlapping miRNAs are listed in (S5 Table in S1 File).

## 3.4. Expression pattern of miR-1183 at tumor stages

To investigate the expression pattern of miR-1183 during tumor stages (T-stage), the T-stage information was extracted from CRC1 and CRC2. We found that hsa-miR-1183 expression in CRC tissues was considerably upregulated compared with corresponding noncancerous tissues (p $\leq$ 0.05). High expression of miR-1183 is evident in CRC T-stages (T1-T4). However, no regular pattern of miRNA expression in T1-T4 stages was revealed (S3 Fig). In both CRC1 and CRC2, similar patterns were observed showing the significant upregulation of miR-1183 from T1 to T3 stages.

## 3.5. Target genes and their expression correlation with miR-1183

Generally, miRNAs exert their function by binding to the 3' UTR of target mRNA. Therefore, to identify the best possible target(s) of miR-1183, microarray expression data of 52 CRC patients was analyzed. For this purpose, a total 41,077 genes, from mRNA microarray data, were scrutinized to identify genes showing negative correlation (p $\leq$ 0.05 and r $\leq$ -0.5) to miR-1183 expression. There was total 43 genes showing an inverse relationship with miR-1183 (S6 Table in S1 File). Of note, we also validated these putative targets using Target Scan database and miRmap database. According to the results, five genes (*PJA2*, *CRIM1*, *PNRC1*, *CCPG1*, and *TMEM136*) were overlapping containing the seed sequence for miR-1183 (Fig 2E and S7 Table in S1 File). Out of these, *CCPG1* and *CRIM1* turned out to be strong candidates as miR-1183 target (Fig 3A and 3B). Function of the target genes were looked up in NCBI database (S6 Table in S1 File). Out of these genes, cell cycle progression gene 1 (*CCPG1*) gene was found to be involved in the cell cycle. It could have a biological functional effect on the upregulation of miR-1183.

## 3.6. Cell Cycle Progression Gene 1 (*CCPG1*), a putative target of miR-1183

*CCPG1* was identified as a target gene for miR-1183. The miRNA binds at the 3'UTR region of the gene to carry out its regulatory activity (Fig 4A). To investigate the regulatory mechanisms of miR-1183 and *CCPG1* gene, their relative expression in CRC patients was evaluated (Fig 4B). Both show an inverse relationship; miR-1183 is upregulated and *CCPG1* is showing downregulation of expression.

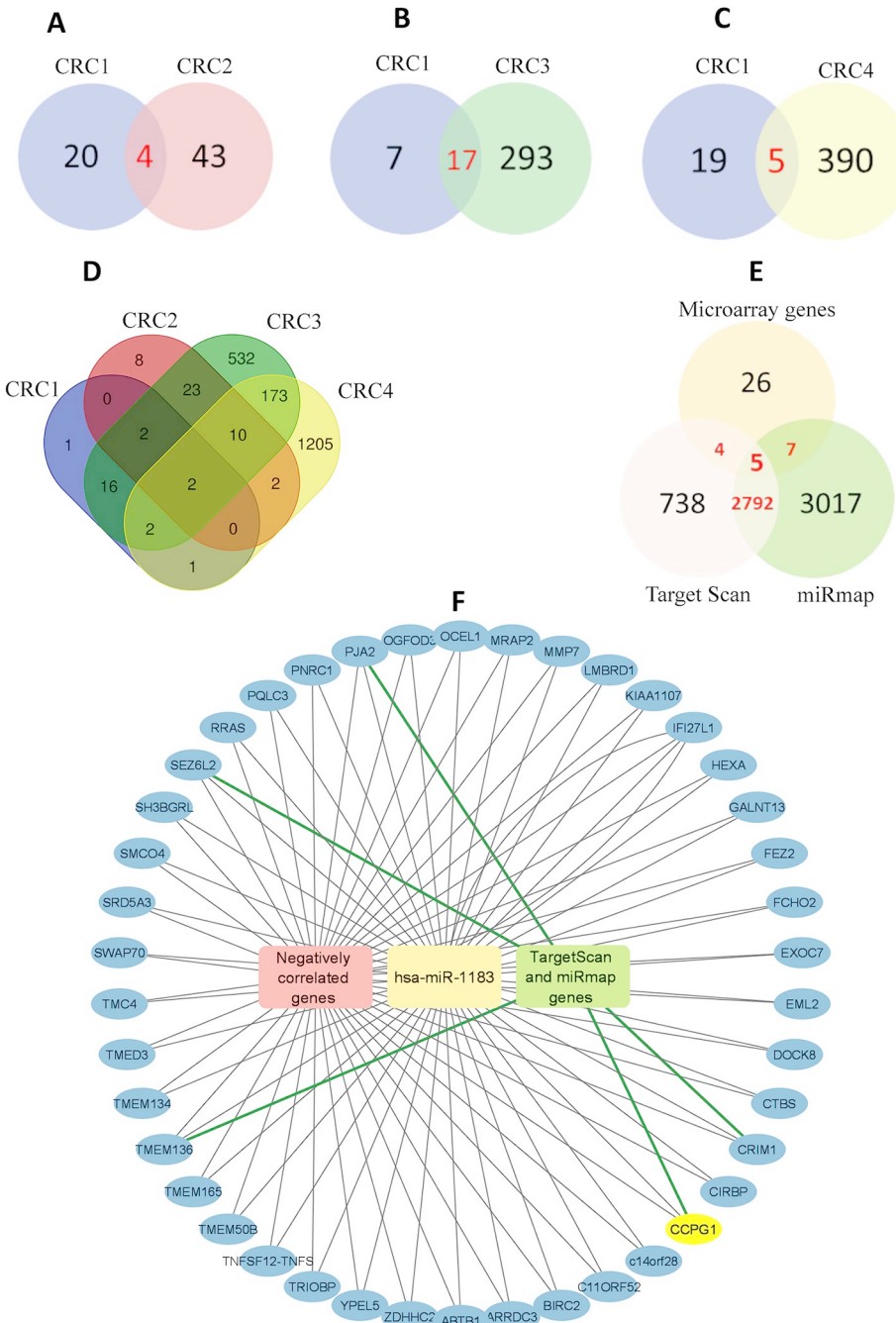

**Fig 2. Common upregulated miRNAs in colorectal cancer datasets.** (A) Common miRNAs between CRC1 and CRC2 (B) Common miRNAs between CRC1 and CRC3 (C) Common miRNAs between CRC1 and CRC4 (D) Venn diagram is showing common miRNAs in colorectal cancer four datasets (CRC1, CRC2, CRC3 and CRC4). (E) Comparison of predicted target genes of miR-1183 from TargetScan, miRmap database, and negatively correlated genes (r ≤ -0.5) from microarray dataset. (F) Network of downregulated genes in CRC that are negatively correlated with hsa-miR-1183 and shared by TargetScan and miRmap.

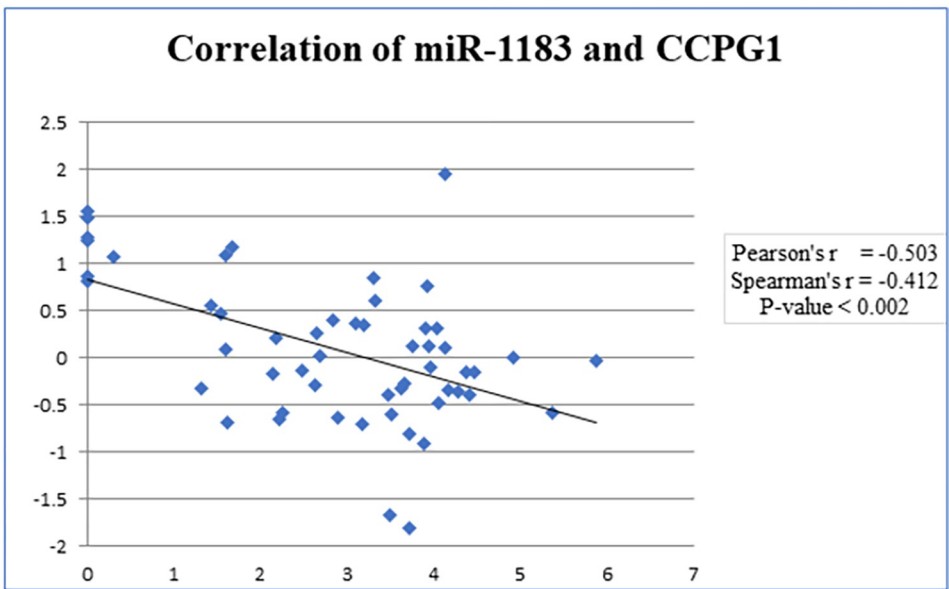

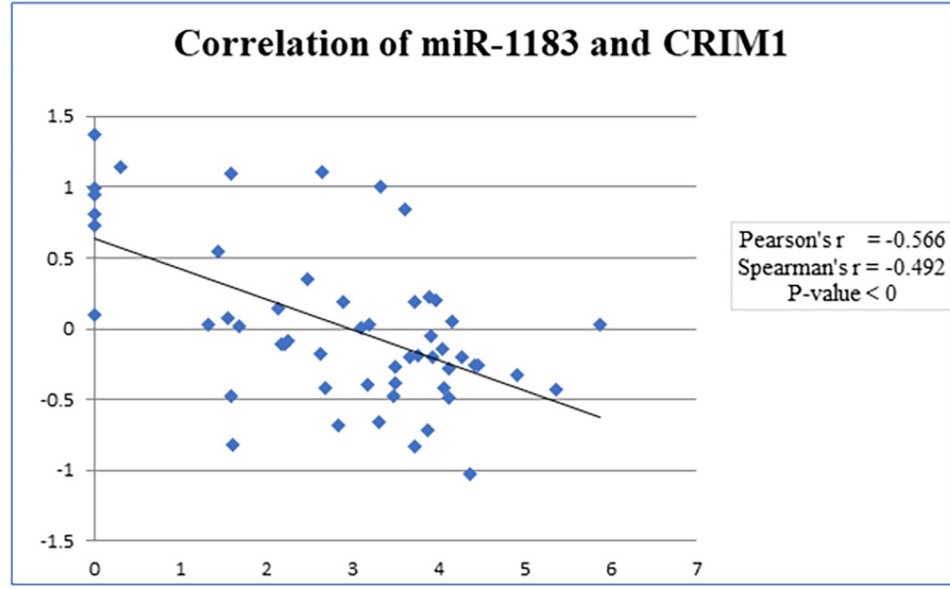

**Fig 3.** (A) *CCPG1* expression is inversely correlated with miR-1183 expression in CRC patients. The X-axis is presenting miR-1183 expression values and Y-axis represents the expression values of *CCPG1*. (B) CRIM1 expression is inversely correlated with miR-1183 expression in CRC patients. The X-axis is presenting miR-1183 expression values and Y-axis represents the expression values of CRIM1.

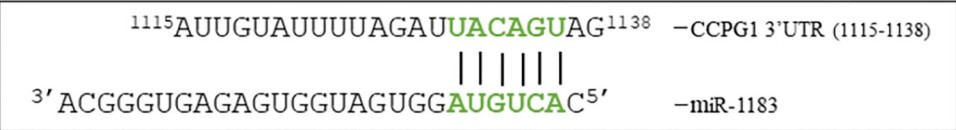

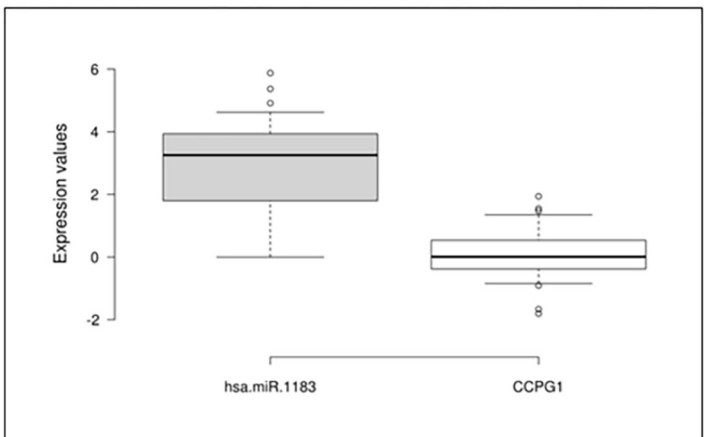

**Fig 4.** (A) Schematic depiction of miR-1183 interaction with 3' ÚTR of *CCPG1*. (B) Boxplot of miR-1183 expression in 50 CRC tissues relative to the expression of *CCPG1*.

## 3.7. Characteristics of samples

The collected samples were verified by histopathologist using microscopy (Fig 5A). The collected patient data was stratified based on clinicopathological criteria such as age, gender, and tumor stages (1–4) (Fig 5B). Number of males diagnosed with colorectal cancer were quite high (78%) as compared to females (22%). This data corroborates with the fact that men are at greater risk of developing colorectal cancer as compared to females [10]. The mean age of patients was 56 years. Furthermore, 87% of the cases were of either stage III or stage IV. This observation again corroborates with the fact that colorectal cancer is generally diagnosed at later stages, and this is the main reason of high mortality rate [10].

## 3.8. Expression validation

Next, expression level of miR-1183 and *CCPG1* was quantified on real time thermocycler. Transcript levels of miR-1183 and *CCPG1* were found to be inversely correlated in CRC patients, with miR-1183 being significantly upregulated ($p < 0.0001$) in tumor samples and *CCPG1* being significantly downregulated ($p < 0.001$) (Fig 6). The expression pattern was similar to the computationally predicted values (Fig 4). The data suggests the potential role of miR-1183 towards colorectal cancer progression.

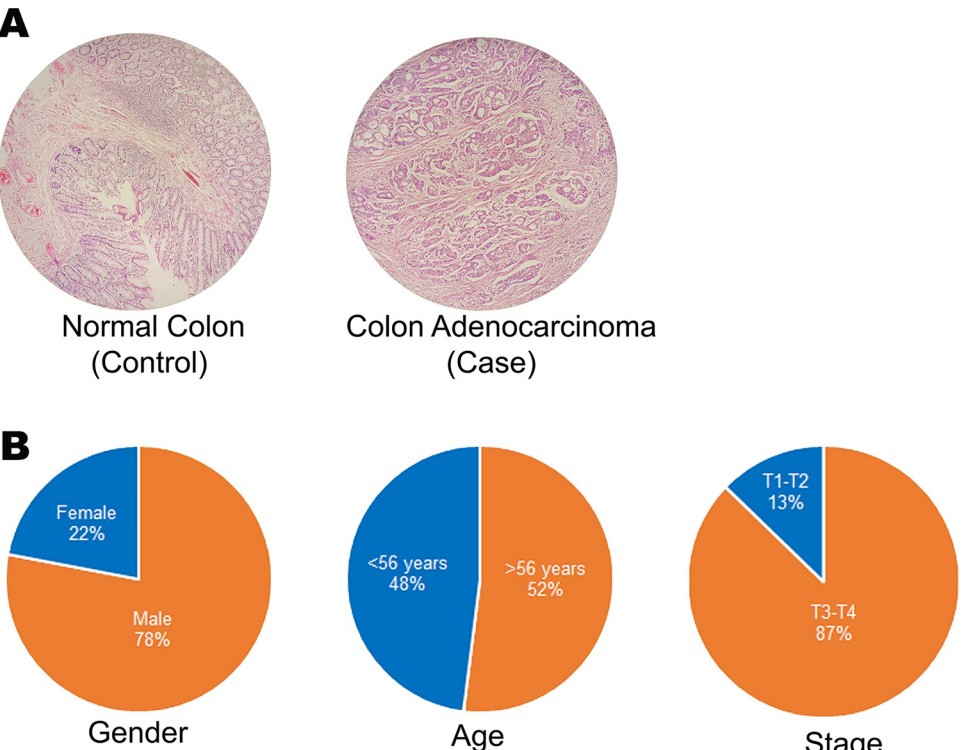

**Fig 5.** (A) Photomicrograph showing normal colon vs colonic adenocarcinoma. (B) Clinical features of 50 CRC patients.

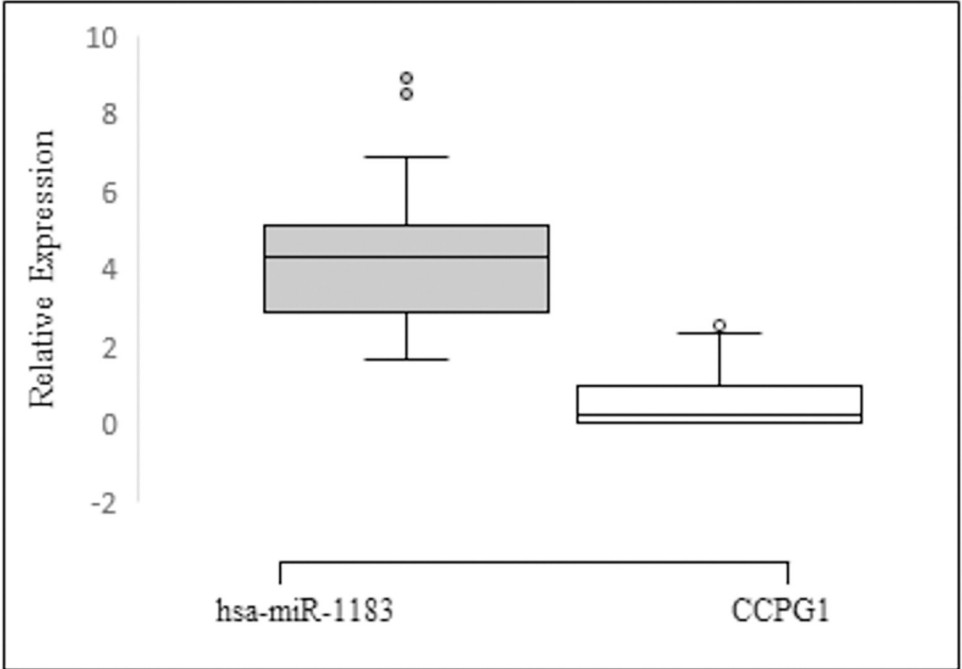

**Fig 6. RT-qPCR analysis of the expression of miR-1183 and *CCPG1*.**

### 3.9. Associated proteins

Since the CCPG1 does not modulate the cell cycle directly [23]. We wanted to understand and know the interacting proteins and the mechanism behind CCPG1 influence on cell cycle. For this purpose, we explored The STRING database, a resource for understanding protein-protein interaction (PPi) networks. The PPi network was constructed using cytoscape with 0.4 (medium) as a cutoff value. We observed that CCPG1 remains in close proximity to RB1CC1 and both are co-expressed. Furthermore, CCPG1 and RB1CC1 had the highest interaction value of 0.979, suggesting a strong link between the two proteins (Fig 7).

### 3.10. Common miRNAs in CRC1, gastric and liver cancer

As metastasis of CRC is evident through literature, the liver being the primary metastatic site of CRC and gastric cancer being is the primary metastatic site of HCC [24], so we were particularly interested in finding out the link between CRC, HCC and GC (Fig 8A). For the purpose, we evaluated the expression pattern of miR-1183 in gastric and liver cancer. Interestingly, miR-1183 was also upregulated in these two cancers, in addition to other miRNAs (Fig 8B and 8C). According to results, five miRNAs including, hsa-miR-630, hsa-miR-1246, hsa-miR-1183, hsa-miR-610 and hsa-miR-650 were common between CRC and HCC patients (Fig 8B). Six miRNAs including, hsa-miR-630, hsa-miR-1246, hsa-miR-1183, hsa-miR-610, hsa-miR-1471 and hsa-miR-650 were common between CRC and gastric cancer patients (Fig 8C), while five miRNAs including, hsa-miR-630, hsa-miR-1246, hsa-miR-1183, hsa-miR-610 and hsa-miR-650 were common among CRC, GC and HCC patients altogether (Fig 8D and 8E).

## Protein-Protein Interactions of CCPG1

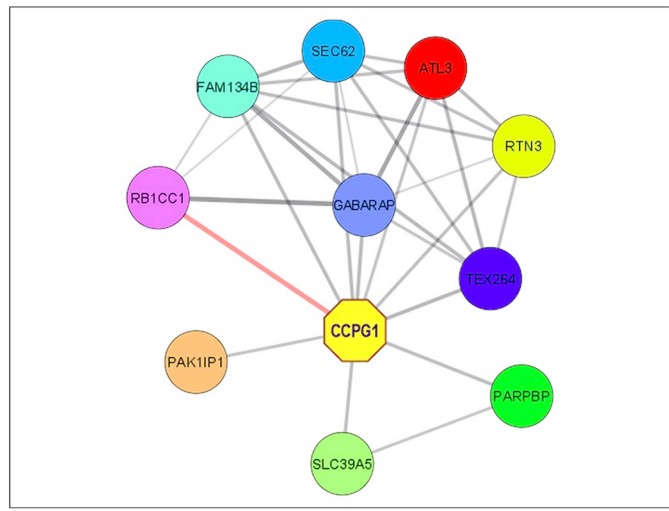

| GENES | RB1CC1 | TEX264 | GABARAP | SEC62 | FAM134B | PARPBP | SLC39A5 | RTN3 | PAK1IP1 | ATL3 |
|---|---|---|---|---|---|---|---|---|---|---|
| SCORE | 0.979 | 0.755 | 0.726 | 0.713 | 0.71 | 0.695 | 0.681 | 0.67 | 0.637 | 0.637 |

**Fig 7. Protein-Protein interactions of CCPG1 generated using STRING database.**

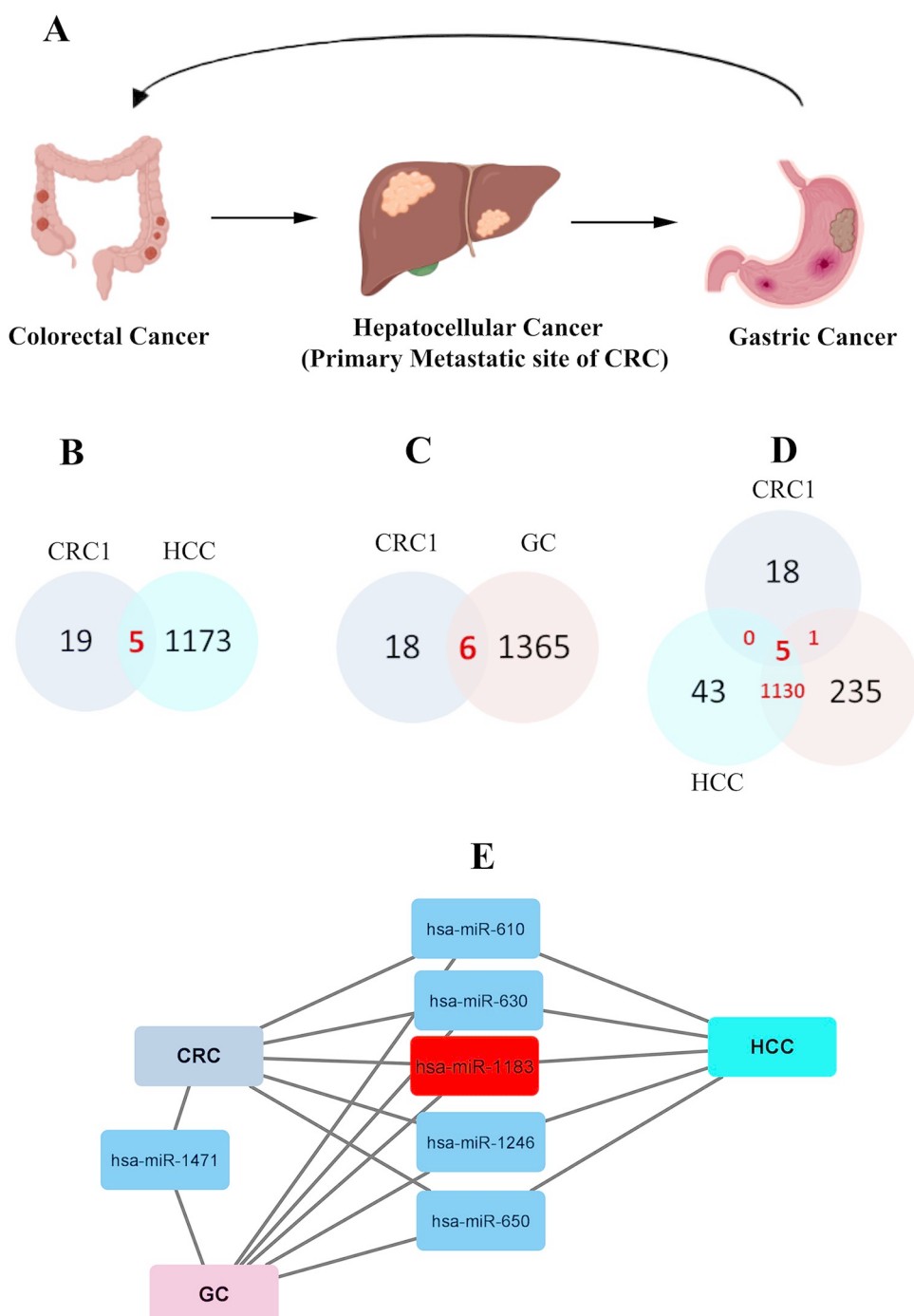

**Fig 8.** (A) Diagrammatic representation of primary metastatic sites for colorectal cancer, liver cacner, and gastric cancer. Venn diagram showing number of common miRNAs in CRC, GC and HCC.(B) Common miRNAs in CRC1 and liver cancer (C) Common miRNAs in CRC1 and gastric cancer (D) Common miRNAs in CRC1, liver, and gastric cancer.(E) Network depiction of five common miRNAs in colorectal cancer, liver cancer, and gastric cancer.

## 4. Discussion

Colorectal cancer (CRC) is one of the most common malignancies, with a high mortality rate [25]. Emerging evidence indicates a critical role of miRNAs in the pathogenesis of cancer

[5,26]. In this study, using publicly available microarray data, we computationally identified miRNAs that have a potential role in CRC progression. We investigated the differential expression of miRNAs that are supposed to be involved in CRC initiation and progression. Our focus was on miRNAs that can act as an oncogene due to upregulated expression. According to the results, CRC dataset 1 (CRC1) yielded 24 miRNAs that were significantly upregulated in CRC comparative to the normal counterparts (S2 Fig). Out of these miRNAs, many are already reported in different cancer types like miR-1182 in gastric and liver cancer [27,28], miR-765 in rectal cancer [29] etc.

Previously, solitary and metachronous metastasis of CRC in HCC [24,30], HCC into GC [31], and of gastric cancer into HCC [32] is reported. In this study, for the first time, we identified CRC related miRNAs and related them with both GC and HCC. Our findings indicate two miRNAs including hsa-miR-630 and hsa-miR-1183 to be significantly upregulated in all three cancer types.

Interestingly, hsa-miR-630 is already reported in many different cancers [33–35]. In all these cancer types, it is reported to be significantly upregulated. The expression of miR-630 is significantly upregulated in lungs cancer [36], HCC [34], gastric cancer [35] and colorectal cancer [33]. Importantly, studies revealed that miR-630 can be used as a potential biomarker to detect CRC [35]. Additionally, studies show the dysregulation of miR-1183 in many different cancers including breast cancer [37] and pancreatic cancer [38]. Recent studies have asserted miR-1183 presence in rectal cancer [29], however their downstream effects, which includes the target genes, and possible association in all three cancers remained unexplored till now. Also, the association of miR-1183 in gastric and liver cancer is largely unknown.

In this study, by analyzing different microarray datasets, we observed that miR-1183 expression was significantly higher in CRC tissues. This study's findings were confirmed by investigating the difference in expression between 50 clinical, and 50 control samples. RT-qPCR results showed a significant (p<0.0001) increase in miR-1183 expression in CRC patients with reference to control.

Expression correlation analysis (Pearson and Spearman) of genes with miRNA was done to find out miRNA potential target genes. Our extensive differential expression and statistical analyses using microarray expression profiling dataset revealed *CRIM1* and *CCPG1* genes to be the potential targets of miR-1183. In particular, the decreased expression of *CRIM1* contributed towards renal cell carcinoma [39]. In contrast, *CRIM1* is upregulated in some cancers like lung cancer [40], and myeloid leukemia [41].

CCPG1 was first discovered as a protein involved in Rho-specific guanine nucleotide exchange. It has binding sites on the PH/DH domain of Dbs (Rho-specific guanine nucleotide exchange factor). The binding of CCPG1 to the Dbs domain limits its activity thus affecting the Rho protein to perform its activity. The downregulation of *CCPG1* in CRC was also observed in an *in vitro* investigation that included 50 case and control samples.

For instance, in retinoblastoma, the most common intraocular tumor, CCPG1 is downregulated by miR-498 [42]. Downregulation of *CCPG1* also affects cell cycle regulation and cause lungs cancer [43]. Moreover, downregulation of *CCPG1* gene has also been reported in colorectal cancer but no experimental work is done yet [44].

By using the string database, we found the association of CCPG1 with RB1CC1 (Fig 7) which acts as a tumor suppressor by downregulating the activity of PYK2 which promotes the proliferation, migration, and invasion of cancer cells [45].

In conclusion, the expression of miR-1183 was predominantly upregulated in CRC and is linked with the progression of tumor. Our findings suggested for the first time, the detailed miR-1183 expression in CRC. This has a potential be used as an independent marker for the prognosis of patients suffering from CRC. Moreover, it is possible that miR-1183 may also

play a significant role in metastasis and invasiveness of CRC. miR-1183 and *CCPG1* both, may serve as prognostic markers in various clinical practices. Further studies carried out to validate the molecular mechanism by which miR-1183 and *CCPG1* carry out their role in different cancer would be highly useful especially for the discovery of an effective therapy against CRC.

## Supporting information

**S1 Fig. Heatmap of differentially expressed miRNAs in colorectal cancer dataset GSE41655 (CRC1).**
(TIF)

**S2 Fig. Significantly dysregulated miRNAs in colorectal cancer dataset (CRC1).**
(TIF)

**S3 Fig. Expression values of miR-1183 in colorectal cancer datasets (CRC1 and CRC2).**
(TIF)

**S1 File. Supplementary tables.**
(DOCX)

## Acknowledgments

The authors would like to acknowledge the conducive environment provided by COMSATS University Islamabad to carry out this research work. They would also like to thank Mr. Afraz Ahmad for his input in improving the readability of the manuscript by proofreading.

## Author Contributions

**Conceptualization:** Farhan Haq, Hassaan Mehboob Awan.

**Formal analysis:** Syeda Alina Fatima.

**Funding acquisition:** Hassaan Mehboob Awan.

**Investigation:** Syeda Alina Fatima, Mubeen Tabish Nasim.

**Methodology:** Ambrin Malik.

**Project administration:** Hassaan Mehboob Awan.

**Resources:** Saboora Waris, Manal Rauf, Syed Salman Ali.

**Supervision:** Farhan Haq, Hassaan Mehboob Awan.

**Writing – original draft:** Syeda Alina Fatima, Mubeen Tabish Nasim.

**Writing – review & editing:** Saif Ur Rehman, Hassaan Mehboob Awan.

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
