## [Decision Letter · Decision Letter 0]

12 May 2023

PONE-D-23-09843In silico and expression analysis shows miR-1183 to be upregulated in colorectal cancer and targets cell cycle progression gene 1PLOS ONE

Dear Dr. Awan,

Thank you for submitting your manuscript to PLOS ONE. After careful consideration, we feel that it has merit but does not fully meet PLOS ONE’s publication criteria as it currently stands. Therefore, we invite you to submit a revised version of the manuscript that addresses the points raised during the review process.

We look forward to receiving your revised manuscript.

Kind regards,

Abdul Rauf Shakoori

Academic Editor

PLOS ONE

Journal Requirements:

Reviewers' comments:

Reviewer's Responses to Questions

**Comments to the Author**

1. Is the manuscript technically sound, and do the data support the conclusions?

Reviewer #1: Partly

Reviewer #2: Partly

2. Has the statistical analysis been performed appropriately and rigorously? 

Reviewer #1: Yes

Reviewer #2: No

3. Have the authors made all data underlying the findings in their manuscript fully available?

Reviewer #1: Yes

Reviewer #2: Yes

4. Is the manuscript presented in an intelligible fashion and written in standard English?

Reviewer #1: No

Reviewer #2: No

5. Review Comments to the Author

Reviewer #1: Dear Editor,

After reading the submitted manuscript entitled “In silico and expression analysis shows miR-1183 to be up-regulated in colorectal cancer and targets cell cycle progression gene 1”, we noticed that authors used the available expression data in a good rationale and inferred important conclusions related to their work, however, some additional work is required for this manuscript to be accepted for publication:

- The manuscript requires English edition

- Authors claimed that CCPG1 is a miR-1183 target but didn’t give sufficient bioinformatics and experimental

evidences about this relationship. Accordingly, additional work is required to support this suggestion, including: a figure that illustrates the interaction between the miRNA and its target site in the CCPG1 mRNA, in addition to experimental work to determine the impact of ectopic miR-1183 over-expression/down-regulation on the expression of CCPG1 at the mRNA and protein levels. Moreover, Dual Luciferase Assay gives important evidence about this targeting relationship.

Best Regards

Reviewer #2: Fatima et al. proceeded with retrieval of four CRC datasets from GEO and found 2 significant miRNAs (i.e., hsa-miR-1183 and hsa-miR-630) overlapping between all the four CRC datasets. They proceeded with miR-1183 (oncomiR) by identifying its target genes, expression, validation, survival, and PPI analysis. Lastly, they checked for common miRNAs in CRC1, Gastric and liver cancer for establishing metastasis point of view.

1) "Colorectal carcinoma (CRC) is among the foremost malignancies with high mortality rate worldwide (Sung et al., 2005)". The reference mentioned in the introduction section is quite old and needs to be updated as per the latest data.

2) Abbreviations should be mentioned in the first place of their occurrence. The authors have used abbreviated terms and full forms somewhere at other places.

3) The introduction section lacks novelty, strength, and limitations of the study and needs to be explained in detail. A brief paragraph summarizing the study design and results snippets also needs to be incorporated in detail.

4) The abstract section needs to be re-written. Usually the abstract encompasses background, study design, results, conclusions.

5) There are too many typos and grammatical errors throughout the manuscript and needs to be checked by a native English speaker or a professional English editing software.

6) The authors have not mentioned why they retrieved gastric cancer and hepatocellular cancer datasets apart from CRC?

7) The datasets inclusion and exclusion criteria must be properly mentioned in the methods section.

8) The section named "Clinical Information of colorectal cancer (CRC)" in the methods section seems to be irrelevant as it discusses information about CRC datasets. It must be either placed in the supplementary file or as results section.

9) "R-Bioconductor package was used for data normalization". Please mention which package? The authors have not mentioned any information regarding datasets pre-processing which usually involves normalization, log2 transformation, batch correction, gene mapping, etc. The information currently mentioned seems misleading to reader.

10) The authors used only TargetScan for their target search? Why. Please justify. There are other validated sources available also such as miRWalk, miRDB, miRTarBase, etc.

11) It would be nice if the mRNA dataset of CRC also comprised same patients in CRC1/CRC2/CRC3/CRC4 as it would justify the results properly.

12) The KM curve shows a nonsignificant log rank p-value (p-value = 0.33). How can CPG1 is prognostically significant?

13) What cutoff did the authors chose for constructing PPI network from STRING? Please mention.

14) The quality of figures needs to be improved throughout the manuscript.

6. PLOS authors have the option to publish the peer review history of their article (what does this mean?). If published, this will include your full peer review and any attached files.

Reviewer #1: No

Reviewer #2: No

---

## [Author Response · Author response to Decision Letter 0]

26 Jun 2023

1. The manuscript has now been updated according to PLoS One guidelines.

2. Since the samples used in this study were archived ones (year range 2016-2019), informed consent was not possible. Therefore, the data and samples obtained were anonymized and the patient’s name, ID, date of birth, and contact number was hidden. Prior approval was sought from the ethical review board of COMSATS University Islamabad (#CUI/Bio/ERB/2021/53). The approval letter has been attached with the submission. 

3. Captions for the supporting files has now been included.

4. Response to reviewer 1: Due to lack of infrastrcuture and resources, we were unable to carry out experimental work at this time. However, we have added a figure panel (4A) to show miRNA-target gene interaction. Additionally, we have updated the title to better align with the findings. Thank you for your valuable comments.

5. Response to reviewer 2. We have addressed all the questions asked by you. Thank you for your valuable comments.

Detailed response to reviewer has been uploaded a separate file.

---

## [Editor Report · Decision Letter 1]

11 Jul 2023

In silico analysis and experimental validation shows negative correlation between miR-1183 and cell cycle progression gene 1 expression in oral cancer

PONE-D-23-09843R1

Dear Dr. Awan,

We’re pleased to inform you that your manuscript has been judged scientifically suitable for publication and will be formally accepted for publication once it meets all outstanding technical requirements.

Kind regards,

Abdul Rauf Shakoori

Academic Editor

PLOS ONE
---

## [Editor Report · Acceptance letter]

27 Jul 2023

PONE-D-23-09843R1 

*In silico* analysis and experimental validation shows negative correlation between miR-1183 and cell cycle progression gene 1 expression in colorectal cancer 

Dear Dr. Awan:

I'm pleased to inform you that your manuscript has been deemed suitable for publication in PLOS ONE. Congratulations! Your manuscript is now with our production department. 

Kind regards, 

on behalf of

Dr. Abdul Rauf Shakoori 

Academic Editor

PLOS ONE